# The Prevalence of Multidrug-Resistant *Enterobacteriaceae* among Neonates in Kuwait

**DOI:** 10.3390/diagnostics13081505

**Published:** 2023-04-21

**Authors:** Rehab Zafer Alajmi, Wadha Ahmed Alfouzan, Abu Salim Mustafa

**Affiliations:** 1Department of Microbiology, Faculty of Medicine, Kuwait University, Kuwait City 13110, Kuwait; rehabzalajmi@gmail.com (R.Z.A.);; 2Microbiology Unit, Department of Laboratory Medicine, Farwaniya Hospital, Ministry of Health, Farwaniya 80000, Kuwait

**Keywords:** *Enterobacteriaceae*, drug resistance, resistance genes, neonates, PCR

## Abstract

Increasing numbers of neonates with serious bacterial infections, due to resistant bacteria, are associated with considerable morbidity and mortality rates. The aim of this study was to evaluate the prevalence of drug-resistant *Enterobacteriaceae* in the neonatal population and their mothers in Farwaniya Hospital in Kuwait and to determine the basis of resistance. Rectal screening swabs were taken from 242 mothers and 242 neonates in labor rooms and wards. Identification and sensitivity testing were performed using the VITEK^®^ 2 system. Each isolate flagged with any resistance was subjected to the E-test susceptibility method. The detection of resistance genes was performed by PCR, and the Sanger sequencing method was used to identify mutations. Among 168 samples tested by the E-test method, no MDR *Enterobacteriaceae* were detected among the neonates, while 12 (13.6%) isolates from the mothers’ samples were MDR. ESBL, aminoglycosides, fluoroquinolones, and folate pathway inhibitor resistance genes were detected, while beta-lactam–beta-lactamase inhibitor combinations, carbapenems, and tigecycline resistance genes were not. Our results showed that the prevalence of antibiotic resistance in *Enterobacteriaceae* obtained from neonates in Kuwait is low, and this is encouraging. Furthermore, it is possible to conclude that neonates are acquiring resistance mostly from the environment and after birth but not from their mothers.

## 1. Introduction

*Enterobacteriaceae* is a large and diverse family of Gram-negative bacteria. They vary from existing as a normal flora in the human gut, such as *Escherichia coli*, to presenting as an opportunistic organism causing disease under certain conditions [1]. According to the U.S. National Healthcare Safety Network, 30% of hospital-acquired infections were caused by Gram-negative bacteria, and 70% were the reason behind infections in intensive care units. Among these bacteria, the *Enterobacteriaceae* family are the predominant [2].

Researchers and the healthcare community are facing a global crisis with the emergence and rise of drug resistance conferred by bacteria [3]. According to the global antimicrobial resistance and use surveillance system (GLASS) report published by the World Health Organization (WHO) in 2021, antimicrobial resistance is one of the top ten global public health concerns that threaten humans [4]. Furthermore, WHO has declared carbapenem-resistant and third-generation cephalosporin-resistant *Enterobacteriaceae* as a critical priority for research and development [5].

Multi-antibiotic resistance in *Enterobacteriaceae* is an increasing problem, with the strains being resistant to most available antibiotics [6]. Increasing numbers of neonates with serious bacterial infections, due to resistant bacteria, are associated with considerable morbidity and mortality rates. A global report published in 2013 revealed that 6.3 million live-born children died before the age of 5 years worldwide, and half of them died of infectious causes, with neonatal sepsis as the third leading cause, accounting for 15 % of the total infection-related under-5-year-old child deaths [7].

β-lactams, fluoroquinolones, and aminoglycosides are the three main groups of antibiotics that *Enterobacteriaceae* confer resistance to, in addition to folate pathway inhibitors (trimethoprim-sulfamethoxazole) and tigecycline from the glycylcycline group. In general, bacteria follow more than one mechanism to resist antibiotics, including (1) enzymatic modification of the drug, (2) modification of the structure of the target, (3) efflux pumps, and (4) reduction in the penetration of the drug due to changes in cell wall permeability [8]. The mechanism of resistance of each group of antibiotics is explained below.

Three mechanisms of resistance to β-lactams are commonly exhibited by *Enterobacteriaceae*, which include the production of enzymes, porin defects, and efflux pump overexpression. The production of inactivating enzymes is predominant. These enzymes, named β-lactamases, hydrolyze the β-lactam ring and inactivate the antibiotic. They fall into four groups, A, B, C and D, according to their molecular classification. Class A show resistance to penicillins and early cephalosporins, while class B, the metallo-β-lactamases, confer resistance to carbapenems. Class C, the chromosomal AmpC β-lactamases, confer resistance to third-generation cephalosporins and β-lactamase inhibitors, and Class D, the oxacillinases, show resistance varying from narrow-spectrum antibiotics, including penicillin and the first generation of cephalosporins, to extended-spectrum antibiotics, including late-generation cephalosporins [6,9,10]. Extended-spectrum β-lactamase (ESBL)-producing *Enterobacteriaceae* produce enzymes belonging mostly to Ambler class A of the β-lactamases, which hydrolyze most penicillins and cephalosporins (third- and fourth-generation) but not cephamycins or carbapenems. These enzymes are inhibited by β-lactamase inhibitors including clavulanic acid, sulbactam, tazobactam, and avibactam [11]. They are introduced in combination with β-lactams to decrease the activity of β-lactamases [12]. Enzymatic mechanisms are the main cause of resistance to β-lactamase inhibitors. Furthermore, mutations at critical sites in β-lactamases such as TEM, SHV, KPC, and CTX-M lead to resistance to β-lactam/β-lactamase inhibitor combinations [13,14].

Several mechanisms of resistance to aminoglycosides have been identified within *Enterobacteriaceae*, including the modification of the drug, decreased uptake of the drug because of membrane impermeabilization, modification of the target, which is the 30S ribosomal subunit, and an increased efflux pump [15]. The common mechanism of resistance among *Enterobacteriaceae* is the enzymatic modification of aminoglycosides, which is mediated by aminoglycoside-modifying enzymes (AMEs). The AMEs are N-Acetyltransferases (AAC), O-Adenyltransferases (ANT), and O-Phosphotransferases (APH). The ACC catalyzes the acetyl CoA-dependent acetylation of an amino group, while ANT catalyzes the ATP-dependent adenylation of a hydroxyl group, and APH catalyzes the ATP-dependent phosphorylation of a hydroxyl group [16].

Bacteria may exhibit two general mechanisms of resistance to fluoroquinolones. The first is based on mutations occurring in two classes of target enzymes. In the case of Gram-negative bacteria, mutations leading to resistance to fluoroquinolones occur in the subunits of GyrA, while mutations in ParC topoisomerase IV tend to play a secondary role. The second general mechanism is the use of efflux pumps. These may continuously remove the antibiotic outside the cytosol, leading to decreased susceptibility to the drug. Furthermore, the outer membrane of Gram-negatives is composed of a decreased level of diffusion porins, resulting in reduced accumulation of the drug inside the cell [17].

The mechanism of resistance to tigecycline is mainly due to efflux pumps. It has been suggested that resistance may occur due to the acquisition of *tetX* genes, which are similarly found in tetracycline-resistant *Enterobacteriaceae* [18]. Additionally, it may occur as a result of AcrAB efflux pump overexpression due to mutations [19].

Several mechanisms function to resist trimethoprim-sulfamethoxazole. The acquisition of genes that code isoforms of dihydropteroate-synthase- and dihydrofolate-reductase-targeted enzymes confers resistance to sulfonamides and trimethoprim, such as *sul1* and *sul2*, and many variants of *dfr* genes [20,21]. Additionally, modifications of the targeted enzymes occur due to mutations in the encoding genes. As a regulated cycle, the folate synthesis cycle is affected by any changes in enzymes, especially the overexpression of enzymes, which can lead to resistance in bacteria [22].

The standardization of the pattern of resistance to antibiotics may show resistance trends and thus provide guidelines for their safer use. The Clinical Laboratory Standards Institute (CLSI) [23] and other institutes define antibiotic resistance categories using data collected based on minimum inhibitory concentration testing and the disk diffusion method. Three types of resistance are known, which are multidrug resistance (MDR), resistance to at least three different classes of antibiotics; extensive drug resistance (XDR), resistance to all but two classes of antibiotics; and pan-drug resistance (PDR), resistance to all classes of antibiotics [24].

The objective of this work was to evaluate the prevalence of multi-drug-resistant *Enterobacteriaceae* in the neonatal population and their mothers in Farwaniya Hospital in Kuwait and to determine the basis of resistance by polymerase chain reaction (PCR) and partial gene sequencing.

## 2. Materials and Methods

### 2.1. Sample Collection

Samples were obtained from the Obstetrics and Gynecology labor rooms and wards of Farwaniya Hospital. A total of 484 rectal swabs were collected, including 242 samples from neonates and 242 samples from the mothers. All the neonate samples were collected from full-term babies. Each sample was labeled with a different number. The mother and the neonate samples were labeled with the same number, with the addition of (B/O) the neonates’ samples. The exclusion criteria included neonates older than one day, missed samples (one swab collected for the neonate only or mother only), and unlabeled samples. Ethical approval for the study was obtained from the Ministry of Health of Kuwait, Asst. Undersecretary for Planning and Quality (Research number 1211/2019). A research consent form that explained the research was signed for each mother and baby participating in this project, as shown in Appendix A.

### 2.2. Isolation and Identification

*Enterobacteriaceae* species were isolated from the collected rectal swabs by culturing the specimens on MacConkey agar and MacConkey agar supplemented with meropenem at a concentration of 1 µg/mL. The plates were incubated at 37 °C for 24 h. Identification and sensitivity testing were performed by running a bacterial suspension with turbidity equivalent to 0.5 McFarland standard of each isolate in a VITEK^®^ 2 system (bioMérieux, MarcyL’Étoile, France). The isolates were stored in a preservative medium containing brain heart infusion broth, distilled water, and glycerol at −70 °C until further processing.

### 2.3. Susceptibility Test by E-Test Method

Each isolate flagged in the VITEK^®^ 2 system results with any of the terms ESBL, aminoglycosides, and fluoroquinolones was tested for susceptibility by the E-test method. Bacterial suspensions with turbidity equivalent to 0.5 McFarland standard were spread onto Mueller–Hinton agar to determine the minimum inhibitory concentration (MIC) of individual antimicrobial agents according to CLSI for the E-test method (Marcy-l’Étoile—France). The antibiotics used for the E-test are shown in Table 1. The plates were incubated at 37 °C for 24 h. The control strain used in this method was *Escherichia coli* ATCC 25922.

### 2.4. Molecular Detection of Resistance Genes

DNA was extracted using the QIAamp DNA Mini Kit 50 (QIAGEN, Hilden, Germany), following the manufacturer’s instructions. Molecular detection of each resistance gene for the different antibiotic groups was performed by polymerase chain reaction (PCR); the primers used are given in Appendix A. The PCR reaction mixture (a total of 25 μL) contained 2 μL of extracted DNA as a template, 1 μL of forward primer, 1 μL of reverse primer, 4 μL of 5× hot firepol^®^ blend master mix ready to load (Solis BioDyne, Tartu, Estonia), and 17 μL nuclease-free water. For the detection of each group of antibiotics, a different PCR cycle was performed, as shown in Table 2. Agarose gel electrophoresis was applied to observe the bands of DNA.

### 2.5. Sequencing of Resistance Genes

For the detection of mutations in the genes (*gyrA*, *parC*), Sanger sequencing of the target genes was performed. The DNA extracted from each bacterial isolate was amplified using the primers given in Appendix A. The reaction setup was as follows: a total volume of 25 μL reaction mixture in a PCR tube containing 10 μL AmpliTaq Gold Master Mix, 0.5 μL forward primer (10 μM), 0.5 μL reverse primer (10 μM), 2 μL template DNA, and 12 μL nuclease-free water. The DNA targets were amplified using a Gene Amp PCR System 9700 under the following conditions: initial denaturation step of 12 min at 95 °C followed by 30 amplification cycles of denaturation at 94 °C for 1 min, annealing at 57 °C for 1 min, extension at 72 °C for 1 min, and a final extension step at 72 °C for 10 min. The PCR product was purified using the Exosap Purification kit. The sequencing PCR was performed with the purified PCR product and BigDye Terminator v1.1 Cycle Sequencing kit (Applied Biosystems, Grand Island, NY, USA). Using the Gene Amp PCR system 9700, the DNA was amplified under the following conditions: denaturation at 96 °C for 1 min followed by 25 cycles of 96 °C for 10 s, annealing at 50 °C for 5 s, and extension at 60 °C for 4 min. The sequencing PCR product was purified using the BDX Terminator Purification Kit (Applied Biosystems, Grand Island, NY, USA), according to the manufacturer’s instructions. In brief, 10 μL of sequencing PCR product was added to 10 μL of BDX Terminator solution and 45 μL of SAM solution (Applied Biosystems, Grand Island, NY, USA). The mixture was placed in a 96-well plate and sealed with a ‘Microseal’ adhesive film. The contents were mixed using a plate vortex for half an hour. The plate was centrifuged at 1000 rpm for 2 min. The sequencing plate was fixed in an adaptor and kept in the ABI 3130 Genetic Analyzer for sequence determination. The 1X running buffer was fixed to the red line in the anode buffer jar and to the black line in the buffer trough position no. 1. The sample information, analysis protocol results, and the group and instrument protocol were entered into the plate manager ID sheet. The sample plate was loaded onto one of the decks, and the plate run ID was linked to the plate. Then, the sequencing run was started.

### 2.6. Construction of Phylogenetic Tree

A phylogenetic tree was constructed based on DNA gyrase A (*gyrA*) and DNA topoisomerase IV (*parC*) sequences using MEGA11 software to illustrate the evolutionary distance between the isolates that showed mutations and *Escherichia coli* strains with known pathotypes, which were obtained from National Center for Biotechnology Information/Nucleotide BLAST.

## 3. Results

### 3.1. Isolation and Identification

During the period from January to April 2020, 484 samples were collected (242 samples from neonates and 242 samples from mothers). A total of 328 *Enterobacteriaceae* species were isolated from 484 samples. These species included *Escherichia coli* (n = 232, 70.7%), *Klebsiella pneumoniae* (n = 48, 14.6%), *Klebsiella oxytoca* (n = 3, 0.9%), *Klebsiella ozaenae* (n = 1, 0.3%), *Enterobacter cloacae* (n= 18, 5.5%), *Enterobacter aerogenes* (n = 1, 0.3%), *Citrobacter freundii* (n = 10, 3%), *Citrobacter farmeri* (n = 5, 1.5 %), *Citrobacter braakii* (n = 2, 0.6%), *Citrobacter koseri* (n = 1, 0.3%), *Citrobacter amalonaticus* (n = 1, 0.3%), *Kluyvera ascorbata* (n = 4, 1.2%), *Shigella sonnei* (n = 1, 0.3%), and *Cronobacter sakazakii* (n = 1, 0.3%), as shown in Table 3. The sensitivity testing was performed using the VITEK^®^ 2 system with 328 Enterobacteriaceae isolates. Out of 44 neonates’ isolates, 4 isolates were flagged as resistant, and all were ESBL-positive. On the other hand, out of 284 isolates obtained from the mothers’ samples, 35 isolates were flagged as resistant, of which 33 isolates were ESBLs, 2 were resistant to aminoglycosides, and 2 were both ESBLs and showed aminoglycosides resistant.

### 3.2. Susceptibility Test by E-Test Method

A total of 88 isolates obtained from the mothers’ samples were tested for susceptibility by the E-test method, while from neonates, only 12 isolates were tested for susceptibility by the E-test method. In the antibiotic susceptibility testing by the E-test method for the neonates’ isolates, only nine isolates revealed resistance to at least one antibiotic, and none of them were multidrug-resistant. Meanwhile from the mothers’ isolates, 55 isolates revealed resistance to at least 1 group of antibiotics. Among the resistant isolates, 13.6 % were multidrug-resistant, with resistance to at least three different groups. The numbers of isolates resistant to each antibiotic among the mothers and neonates are shown in Table 4.

### 3.3. Molecular Detection of Resistance Genes

Out of 44 isolates tested for the presence of ESBL genes, 33 carried the gene blaCTX-M, 5 carried the gene blaTEM, and 2 carried the gene blaSHV. Out of 35 isolates resistant to trimethoprim-sulfamethoxazole, 30 isolates carried the gene sul2, 23 isolates carried sul1, and 4 isolates carried sul3, while 25 isolates carried the gene dfr7&17, 7 isolates carried dfr1, and 6 isolates carried dfr5. Out of five isolates resistant to aminoglycosides, four isolates carried the gene aac(3)-II, two isolates carried aac(6′)-Ib, two isolates carried aac(6′)-II, and two isolates carried the gene ant(3″)-I. Out of 15 isolates resistant to ciprofloxacin, 3 isolates carried the qnrS gene, 2 isolates carried qnrA, and 1 carried qnrB. Resistance genes of amoxicillin-clavulanic acid, carbapenems, and tigecycline were not detected.

### 3.4. Sequencing of Resistance Genes

Mutations were detected in both the gyrA and parC genes. Out of 16 isolates tested, 14 isolates revealed mutations. Alteration from serine to leucine at position 83 in gyrA was detected in 14 isolates, while alteration from aspartate to asparagine at position 87 in gyrA was detected in 13 isolates. Furthermore, alteration from serine to isoleucine at position 80 in the parC gene was detected in 13 isolates.

### 3.5. Construction of Phylogenetic Tree

Phylogenetic trees based on DNA gyrase A (gyrA) and DNA topoisomerase IV (parC) sequences illustrating the evolutionary distance between 16 isolates of Escherichia coli from 14 mothers and 2 neonate samples and strains of Escherichia coli with known pathotypes are shown below in (Figure 1) and (Figure 2).

## 4. Discussion

To our knowledge, this was the first study performed in Kuwait regarding the prevalence of MDR *Enterobacteriaceae* among neonates. *Escherichia coli* and *Klebsiella pneumoniae* were the *Enterobacteriaceae* species most frequently isolated from both the mothers’ and neonates’ samples in this study. These species are known to be the most frequent Gram-negative bacteria causing infections in hospitals and the community according to the World Health Organization [30].

Among the tested samples, 13.6% were MDR (n = 12), and all of them were isolated from mothers. Furthermore, all the MDR *Enterobacteriaceae* species were *Escherichia coli*. None of the isolates obtained from the neonates’ samples were MDR. Only six of the neonates’ samples conferred resistance in the same way as the mother. Among 12 MDR samples from mothers, only two samples conferred resistance to their neonates, and both were *Escherichia coli*. This study further shows that the mothers’ samples had a higher resistance rate compared to those of neonates. This indicates that neonates’ resistance was not acquired from mothers, and it might be acquired later from the hospital setting or community.

Multidrug resistance among neonates, especially in neonatal ICUs, is a matter of great concern. Based on studies conducted globally, ESBL- and carbapenem-resistant *Enterobacteriaceae* are the reasons for NICU outbreaks [31]. Although this study showed a very low rate of resistance in neonates, it is still an increasing threat worldwide. The European Project on Antibiotic Resistance and Prescription in Children (ARPEC) revealed that the rate of resistance to third-generation cephalosporins among Enterobacteriaceae was 6.6–39% [32].

In cases of excessive or unregulated use of antibiotics, commensal bacteria may become reservoirs of antibiotic resistance genes that may later be transferred to pathogenic bacteria [33]. Several studies were conducted regarding resistant commensal *Escherichia coli* among children and neonates, resulting in a high prevalence of resistance [33,34]. A study performed in Ado-Ekiti, Nigeria, aimed to determine the carriage of antibiotic-resistant commensal *Escherichia coli* among infants aged below 5 months. It was found that 48.1 % of the total 212 samples collected from neonates showed growth of *Escherichia coli*. Moreover, 75 % of the isolated *Escherichia coli* were multiple-resistant to three or more antibiotics [33]. Furthermore, a study performed in Vietnam aimed to investigate the prevalence of resistance in commensal *Escherichia coli* obtained from 818 children aged from 6 to 60 months and concluded a high prevalence of resistance among preschool children in rural Vietnam. In total, 60% of the isolates were resistant to three or more antibiotics [34].

According to the definition of ESBL, only 3 isolates obtained from neonates were ESBL producers, while 37 ESBL-producing isolates obtained from the mothers. A high prevalence of ESBL-producing *Enterobacteriaceae* was also demonstrated in another study performed in Kuwait. In a period of two years, from January 2014 to December 2015, *Enterobacteriaceae* species were isolated from total of 4133 proven cases of blood stream infections, in which 25% of the isolates were ESBL producers and 5.2% were confirmed as carbapenem-resistant. Moreover, 60% of the isolates were *Escherichia coli* [35].

Furthermore, a review was published in 2018 that discussed the epidemiology of common resistant bacterial pathogens in the countries of the Arab League, focusing on third-generation cephalosporin-resistant *Enterobacteriaceae* and carbapenem-resistant Enterobacteriaceae, to name a few. It showed that the prevalence of third-generation cephalosporin-resistant *Enterobacteriaceae* reached 25% among the Gulf Cooperation Council countries, specifically, 25% of 190 isolates in Kuwait, 17% of 629 in Qatar, 7% of 17,895 in Saudi Arabia, and 4% of 150 in Oman [36]. In comparison with this study, we found that the level of resistance was increased, as resistance to third-generation cephalosporin was observed 42% of the tested samples.

The tested genes were chosen based on the resistance revealed by each isolate. Because of the variety of genes conferring resistance to each antibiotic group, only the most common genes were tested.

Every isolate that appeared to be resistant to the third generation of cephalosporins and considered as ESBL, according to the definition, was genotypically tested to detect ESBL genes. The *blaCTX-M* was the most common gene detected among the ESBL genes, with a percentage of 75%. It is known to be dominant among the *blaCTX-M* variants in most regions of the world [37]. The other ESBL genes detected were *blaTEM* (11.3%) and *blaSHV* (4.5%), which mainly confer resistance to early generations of cephalosporins [6]. Ten isolates appeared to be resistant on Mueller–Hinton agar, but the genes were not detected by PCR. This might be because of the presence of other genes encoding β-lactamases that were not tested, such as ampC genes.

Aminoglycoside resistance appeared due to the presence of aminoglycoside-modifying enzymes, nucleotidyltransferases, and acetyltransferases but not due to the presence of phosphotransferases, which are the most common cause of resistance to aminoglycosides [38]. Although modification of the ribosomal target by 16S rRNA methyltransferases is another common mechanism, it was not detected in this study.

Two mechanisms of resistance to fluoroquinolones were tested, including the acquisition of resistance genes and point mutations. It was noticed that mutation was slightly higher in frequency than the acquisition of resistance genes, with 10 isolates carrying resistance genes and 13 showing mutations. The mutation was mainly in the *gyrA* gene, which is more common in Gram-negative bacteria than in *parC* [6].

Similar to ESBL, β-lactamase inhibitor resistance can be a result of ampC genes, which were not tested in this study, and this can explain the resistance detected phenotypically but not by genetic methods. Moreover, resistance to β-lactams is mediated by porin defects and efflux pumps, which were not tested in this study, and this is why resistance appeared phenotypically and not genotypically. Additionally, although it has been shown that resistance to tigecycline from the group of glycylcycline in *Enterobacteriaceae* can occur due to the acquisition of tetX genes [20], these genes were not detected in this study; thus, the results may suggest that resistance was mostly mediated by efflux pumps.

## 5. Conclusions

Multidrug resistance among neonates is a priority in modern medicine. The prevalence of antibiotic resistance in *Enterobacteriaceae* obtained from neonates in Kuwait is low and encouraging, with no MDR *Enterobacteriaceae* species being isolated from neonates and a low percentage of mothers conferring multidrug resistance (only 13.6% of the tested samples). Furthermore, after testing the samples genotypically, it is possible to conclude that neonates are acquiring resistance mostly from the environment and after birth but not from their mothers. Further studies including samples from other hospitals in Kuwait would be helpful for the evaluation of the status of antibiotic resistance among *Enterobacteriaceae* species isolated from neonates. Whole-genome sequencing could prove useful for the determination of resistance-conferring mutations that were not detected by PCR and Sanger sequencing in the present study.

## Figures and Tables

**Figure 1 diagnostics-13-01505-f001:**
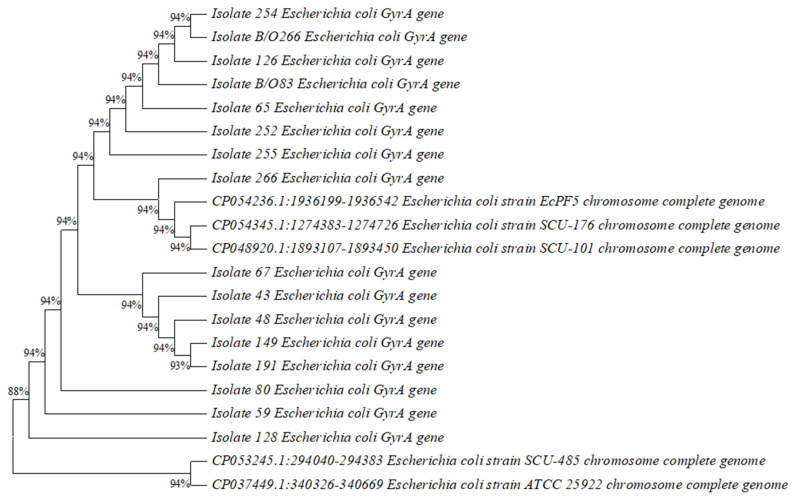
Phylogenetic tree based on DNA gyrase A (*gyr A*) sequences illustrating the evolutionary distance between 16 isolates of *Escherichia coli* from 14 mothers and 2 neonates and 5 strains of *Escherichia coli* with known pathotypes.

**Figure 2 diagnostics-13-01505-f002:**
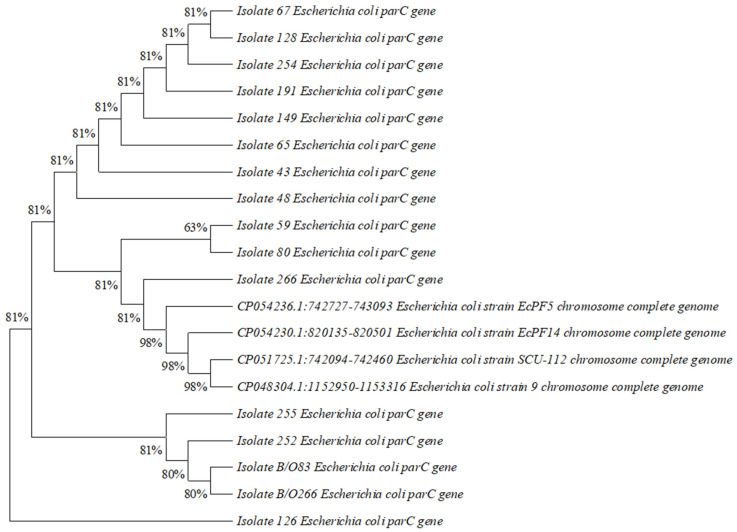
Phylogenetic tree based on DNA topoisomerase IV (*parC*) sequences illustrating the evolutionary distance between 16 isolates of *Escherichia coli* from 14 mothers and 2 neonates and 4 strains of *Escherichia coli* with known pathotypes.

**Table 1 diagnostics-13-01505-t001:** The antibiotics used for the E-test and their concentrations.

Antibiotic Name	Concentration μg/mL
Amoxicillin-clavulanic acid	0.016–256 μg/mL
Ceftolozane-tazobactam	0.016–256 μg/mL
Cefazolin	0.016–256 μg/mL
Cefepime	0.016–256 μg/mL
Ceftaroline	0.016–256 μg/mL
Cefotaxime	0.016–256 μg/mL
Ceftriaxone	0.016–256 μg/mL
Cefoxitin	0.016–256 μg/mL
Cefixime	0.016–256 μg/mL
Ceftazidime	0.016–256 μg/mL
Ceftazidime-avibactam	0.016–256 μg/mL
Ertapenem	0.002–32 μg/mL
Imipenem	0.002–32 μg/mL
Meropenem	0.002–32 μg/mL
Amikacin	0.016–256 μg/mL
Gentamicin	0.064–1024 μg/mL
Ciprofloxacin	0.002–32 μg/mL
Trimethoprim-sulfamethoxazole	0.002–32 μg/mL
Tigecycline	0.016–256 μg/mL

**Table 2 diagnostics-13-01505-t002:** The PCR cycles performed for each antibiotic group.

Antibiotic Group	Gene	Denaturation	Annealing	Extension	Reference
β-lactam-β-lactamase inhibitor combinations (amoxicillin-clavulanic acid)	*bla-TEM, bla-SHV, bla-OXA-1, bla-OXA-2, bla-OXA-10*	95 °C for 1 min/30 cycles	55 °C for 1 min	72 °C for 1 min	[13]
ESBL’s genes	*bla-CTX-M, bla-TEM, bla-SHV*	95 °C for 30 s/35 cycles	60 °C for 30 s	72 °C for 1 min	[25]
Carbapenems	*bla-IMP, bla-VIM, bla-OXA-48, bla-GIM, bla-KPC, bla-NDM*	94 °C for 30 s/36 cycles	52 °C for 40 s	72 °C for 50 s	[26]
Aminoglycosides	*aac(3)-II, aac(6′)-Ib, aac(6′)-II, ant(3″)-I, aph(3′)-VI, armA, rmtB*	94 °C for 30 s/30 cycles	56 °C for 30 s	72 °C for 1 min	[27]
Fluoroquinolones	*qnrA, qnrB, qnrS, qepA, aac(6)-Ib-cr*	94 °C for 1 min/30 cycles	57 °C for 1 min	72 °C for 1 min	[28]
Folate pathway inhibitor (trimethoprim-sulfamethoxazole)	*sul1, sul2, sul3, dfr1, dfr5, dfr7&17*	94 °C for 1 min/30 cycles	57 °C for 1 min	72 °C for 1 min	[20,21]
Tigecycline	*tetX, tetX2, tetX3, tetX4, tetX5*	95 °C for 30 s/30 cycles	58 °C for 30 s	72 °C for 30 s	[29]

**Table 3 diagnostics-13-01505-t003:** The distribution of species among mothers and neonates.

Name of the *Enterobacteriaceae* Isolate	Number Among Mothers	Number Among Neonates
*Escherichia coli*	200	32
*Klebsiella pneumoniae*	39	9
*Klebsiella oxytoca*	3	-
*Klebsiella ozaenae*	1	-
*Enterobacter cloacae*	16	2
*Enterobacter aerogenes*	1	-
*Citrobacter freundii*	10	-
*Citrobacter farmeri*	4	1
*Citrobacter braakii*	2	-
*Citrobacter koseri*	1	-
*Citrobacter amalonaticus*	1	-
*Kluyvera ascorbata*	4	-
*Shigella sonnei*	1	-
*Cronobacter sakazakii*	1	-
Total	284	44

**Table 4 diagnostics-13-01505-t004:** The number of resistant isolates to each antibiotic among mothers and neonates.

Antibiotic Name	Number of Resistant Isolates Among Mothers	Number of Resistant Isolates among Neonates
Amoxicillin-clavulanic acid	36	3
Ceftolozane-tazobactam	34	3
Cefazolin	36	3
Cefepime	29	3
Ceftaroline	29	2
Cefotaxime	34	3
Ceftriaxone	34	3
Cefoxitin	35	3
Cefixime	34	3
Ceftazidime	34	3
Ceftazidime-avibactam	-	3
Ertapenem	2	-
Imipenem	-	-
Meropenem	-	-
Amikacin	1	-
Gentamicin	4	-
Ciprofloxacin	14	2
Trimethoprim-sulfamethoxazole	30	3
Tigecycline	10	2

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
