# Peer review of "The Prevalence of Multidrug-Resistant Enterobacteriaceae among Neonates in Kuwait"

_diagnostics, 2023, doi:10.3390/diagnostics13081505_

Round 1
Reviewer 1 Report
Journal Diagnostics (ISSN 2075-4418)
Manuscript ID
Diagnostics -2308784
Title
The Prevalence of Multi-Drug Resistant Enterobacteriaceae among Neonates in Kuwait
Section
Pathology and Molecular Diagnostics
Special Issue
Diagnosis of Neonatal Diseases
Summery and general comment
The global spread of multidrug-resistant (MDR) Entero-bacteriaceae continues to threaten children worldwide. MDR Enterobacteriaceae can be introduced into the neonates care unit (NCU) environment by colonized children and subsequently spread to other critically ill pediatric patients, resulting in devastating consequences. This is a particular concern with organisms producing extended-spectrum β-lactamases (ESBLs), plasmid-mediated AmpC β-lactamases (pAmpCs), and carbapenemases because genes encoding these enzymes are generally carried on mobile genetic elements, facilitating patient-to-patient transmission. These organisms have been implicated in a number of outbreaks in the NCU setting. Because of frequent transfer to and from acute-care facilities, pediatric patients hospitalized in the NCU may introduce and propagate the spread of resistant pathogens across a variety of healthcare settings. Most reports regarding the molecular characterization of ESBLs, pAmpCs, and carbapenemases in children in the United States focus on clinical isolates. In a study at Seattle Children’s Hospital from 1999 to 2007, 1% of isolates displayed broad-spectrum β-lactamase production, predominantly pAmpCs (blaCMY) and ESBLs (blaTEM). Similarly, 1% of Escherichia coli or Klebsiella spp. isolates recovered from children in Utah from 2003 to 2007 were ESBL producers. Investigators at Texas Children’s Hospital found that ~ 7% of Enterobacteriaceae clinical isolates from 2010 to 2011 were ESBL producers, with CTX-M variants predominating.
General comments
Abstract and Introduction non informative and complex please make them more simple to easy read
Review comment
Highlight all corrections /additions in red color font in revised manuscript.
- All journal names in references must be as per standard journal instruction.
You can check the abbreviation according our journal instruction
- All journal names in references must be as per standard journal instruction. You can check the abbreviation according our journal instruction
- Please add Sequencing trees
Please improved your results
Author Response
Editor in Chief 29th March 13, 2023
Diagnostics
Subject: response to reviewer #1 comments: manuscript ID: diagnostics-2308784- The Prevalence of Multi-Drug Resistant Enterobacteriaceae among Neonates in Kuwait
Thank you very much for sending us the comments of the Reviewer on the above manuscript. We have modified the manuscript in the light of comments and suggestions of the Reviewers. A point-by-point response to the various comments/suggestions of the Reviewers are as follows:
Reviewer #1:
- Comment: Abstract and Introduction non informative and complex please make them more simple to easy read.
- Response: Changes through abstract and introduction have been done.
- Comment: Highlight all corrections /additions in red color font in revised manuscript.
- Response: All corrections/ additions have been colored red.
- Comment: All journal names in references must be as per standard journal instruction. You can check the abbreviation according our journal instruction
- Response: References re-written according to journal instruction.
- Comment: Please add Sequencing trees. Please improved your results.
- Response: Sequencing trees have been added.
Awaiting your positive feedback
Kind regards,
Dr. Wadha A Alfouzan
MB:BS, MSc, FRCPath
Associate Professor of Clinical Microbiology
Department of Microbiology
Faculty of Medicine, Kuwait University
- O. Box 24923, Safat 13110
Kuwait (+965-2498-6516)

Reviewer 2 Report
The manuscript entitled “The Prevalence of Multi-Drug Resistant Enterobacteriaceae among Neonates in Kuwait” provides useful information on the existence and type of antibiotic resistant bacteria in newborns and their mothers. We reckon that it deserves publication after the authors address the issues raised below.
Line 22 “Conclusions: In conclusion, the prevalence of antibiotic resistance…” Delete “In conclusion” and start directly as “The prevalence of antibiotic resistance…”
Lines 30, 170, 256, and throughout the whole text. Names of species should have been written in Italics. Escherichia coli (appearing at least 16 times in text), Klebsiella pneumoniae, Klebsiella oxytoca, Klebsiella ozaenae and all species (including those of table 3 line 267) must be corrected to Italics.
Lines 102-6, you write: “Bacteria mainly exhibit two mechanisms of resistance. The first one is the modifica-tion of the target, which are the two enzymes, by mutations. In Gram-negative bacteria specifically, mutations occur commonly in GyrA subunits rather than ParC topoisomer-ase IV, which tend to play a secondary role in resistance. The second mechanism is the efflux pump, where the cell wall pumps out the drug leading to a low level of resistance.”. Please change to “Bacteria may exhibit two general mechanisms of resistance. The first is based on mutations occurring on two classes of target enzymes. In the case of Gram-negative bacteria, mutations leading to resistance to fluoroquinolones occur in the subunits of GyrA while mutations in ParC topoisomerase IV tend to play a secondary role. The second general mechanism is the use of efflux pumps. These may continuously remove the antibiotic outside the cytosol leading to decreased susceptibility to the drug.”.
Line 116 rimethoprim-sulfamethoxazole. State clearly what is the target enzyme.
Line 117 “Acquisition of genes conferring resistance against sulfonamides and trimethoprim for example sul1 and sul2, and many variants of dfr genes [20]; [21]” What do you mean?
Line 117 “sul1and sul2…”, Line 209 “gyrA, parC …“ and throughout the whole text: all names of all genes should have been written in Italics. Please correct.
Lines 124-26 “It is important to standardize definitions that represent the patterns of resistance to antibiotics to help track the trends of resistance therefore set a suitable method regarding their use.”. This is difficult to understand. Do you mean “The standardization of the pattern of resistance to antibiotics may show resistance trends and thus provide guidelines for their safer use. ”, or something among these lines?
Line 156-157 “meropenem of 1µg/ml”.” What was the point of using the antibiotic at this stage?
Line 156 (“1µg/ml”), 212 “25µl” and throughout all text: Introduce space between numbers and units. You are also advised to do that for % and degrees centigrade.
Line 320-1 “Commensal bacteria are a significant reservoir of genes conferring resistance to antibiotics that can be transferred to pathogenic bacteria [33].”. This statement is confusing and not reflecting the meaning of reference 33. Please change to “In case of excessive or unregulated use of antibiotics, commensal bacteria may become reservoirs of antibiotic resistance genes that may later be transferred to pathogenic bacteria.”.
Line 357 “Ten isolates appeared resistant phenotypically on Muller Hinton agar but the genes.”. Omit “phenotypically”.
Line 377 “indicate” change to “suggest”.
379- 80 “As a global threat with critical priority for research and development, it was important to determine the status of multidrug resistance among neonates.”. Change to “The multidrug resistance among neonates is a priority in modern medicine.”.
386-7 , “…are recommended to evaluate the…”. Change to “would be helpful for the evaluation of …”.
Lines 388-90. You write “For determining the basis of resistance, whole-genome sequencing is recommended since some of the isolates in this study appeared resistant phenotypically, but the genes were not detected by PCR and Sanger sequencing. …”. Change to “Whole genome sequencing could prove useful for the determination of resistance conferring mutations that were not detected by PCR and Sanger sequencing in the present study.”.
Author Response
Editor in Chief 29th March 13, 2023
Diagnostics
Subject: response to reviewer #2 comments: manuscript ID: diagnostics-2308784- The Prevalence of Multi-Drug Resistant Enterobacteriaceae among Neonates in Kuwait
Thank you very much for sending us the comments of the Reviewer on the above manuscript. We have modified the manuscript in the light of comments and suggestions of the Reviewers. A point-by-point response to the various comments/suggestions of the Reviewers are as follows:
Reviewer #2:
- Comment: The manuscript entitled “The Prevalence of Multi-Drug Resistant Enterobacteriaceae among Neonates in Kuwait” provides useful information on the existence and type of antibiotic resistant bacteria in newborns and their mothers. We reckon that it deserves publication after the authors address the issues raised below.
- Response: We thank the reviewer for the positive comment.
- Comment: Line 22 “Conclusions: In conclusion, the prevalence of antibiotic resistance…” Delete “In conclusion” and start directly as “The prevalence of antibiotic resistance…”
- Response: Abstract re-written as suggested.
- Comments: Lines 30, 170, 256, and throughout the whole text. Names of species should have been written in Italics. Escherichia coli (appearing at least 16 times in text), Klebsiella pneumoniae, Klebsiella oxytoca, Klebsiella ozaenae and all species (including those of table 3 line 267) must be corrected to Italics.
- Response: Names of species have been written in Italics.
- Comment: Lines 102-6, you write: “Bacteria mainly exhibit two mechanisms of resistance. The first one is the modification of the target, which are the two enzymes, by mutations. In Gram-negative bacteria specifically, mutations occur commonly in GyrA subunits rather than ParC topoisomerase IV, which tend to play a secondary role in resistance. The second mechanism is the efflux pump, where the cell wall pumps out the drug leading to a low level of resistance.”. Please change to “Bacteria may exhibit two general mechanisms of resistance. The first is based on mutations occurring on two classes of target enzymes. In the case of Gram-negative bacteria, mutations leading to resistance to fluoroquinolones occur in the subunits of GyrA while mutations in ParC topoisomerase IV tend to play a secondary role. The second general mechanism is the use of efflux pumps. These may continuously remove the antibiotic outside the cytosol leading to decreased susceptibility to the drug.”
- Response: Lines 102-6 changed and have been written as suggested.
- Comment: Line 116 trimethoprim-sulfamethoxazole. State clearly what is the target enzyme.
- Response: Line 116 and paragraph bellow have been changed.
- Comment: Line 117 “Acquisition of genes conferring resistance against sulfonamides and trimethoprim for example sul1 and sul2, and many variants of dfr genes [20]; [21]” What do you mean?
- Response: Line 117 have been changed from “Acquisition of genes conferring resistance against sulfonamides and trimethoprim for example sul1 and sul2, and many variants of dfr genes [20]; [21] Also, modifications of the targets which are the enzymes dihydrofolate reductase and dihydropteroate synthase by mutations in the encoding genes.” to “Acquisition of genes that code isoforms of dihydropteroate synthase and dihydrofolate reductase-targeted enzymes, confer resistance against sulfonamides and trimethoprim for example sul1and sul2, and many variants of dfr genes [20]; [21]. Also, modifications of the targeted enzymes by mutations in the encoding genes.” To make it clear.
- Comments: Line 117 “sul1and sul2…”, Line 209 “gyrA, parC … “and throughout the whole text: all names of all genes should have been written in Italics. Please correct
- Response: All names of all genes have been written in Italics.
- Comments: Lines 124-26 “It is important to standardize definitions that represent the patterns of resistance to antibiotics to help track the trends of resistance therefore set a suitable method regarding their use.”. This is difficult to understand. Do you mean “The standardization of the pattern of resistance to antibiotics may show resistance trends and thus provide guidelines for their safer use.”, or something among these lines?
- Response: Yes, the lines 124-26 means “the standardization of the pattern of resistance to antibiotics may show resistance trends and thus provide guidelines for their safer use.” And re-written as suggested.
- Comments: Line 156-157 “meropenem of 1µg/ml”.” What was the point of using the antibiotic at this stage?
- Response: MacConkey agar supplemented with meropenem of 1 µg/ml have been used for primary detection of carbapenem-resistant Enterobacteriaceae.
- Comments: Line 156 (“1µg/ml”), 212 “25µl” and throughout all text: Introduce space between numbers and units. You are also advised to do that for % and degrees centigrade.
- Response: Numbers, units, percentages and degrees re-written as suggested.
- Comments: Line 320-1 “Commensal bacteria are a significant reservoir of genes conferring resistance to antibiotics that can be transferred to pathogenic bacteria [33].”. This statement is confusing and not reflecting the meaning of reference 33. Please change to “In case of excessive or unregulated use of antibiotics, commensal bacteria may become reservoirs of antibiotic resistance genes that may later be transferred to pathogenic bacteria.”
- Response: Line 320-1 re-written and changed as suggested.
- Comments: Line 357 “Ten isolates appeared resistant phenotypically on Muller Hinton agar but the genes.”. Omit “phenotypically”.
- Response: “phenotypically” have been omitted.
- Comments: Line 377 “indicate” change to “suggest”.
- Response: The word has been changed.
- Comments: 379- 80 “As a global threat with critical priority for research and development, it was important to determine the status of multidrug resistance among neonates.”. Change to “The multidrug resistance among neonates is a priority in modern medicine.”.
- Response: Line 379- 80 have been changed as suggested.
- Comments: 386-7, “…are recommended to evaluate the…”. Change to “would be helpful for the evaluation of …”.
- Response: Line 386-7 have been changed as suggested.
- Comments: Lines 388-90. You write “For determining the basis of resistance, whole-genome sequencing is recommended since some of the isolates in this study appeared resistant phenotypically, but the genes were not detected by PCR and Sanger sequencing. …”. Change to “Whole genome sequencing could prove useful for the determination of resistance conferring mutations that were not detected by PCR and Sanger sequencing in the present study.”.
- Response: Line 388-90 have been changed as suggested.
Awaiting your positive feedback
Kind regards,
Dr. Wadha A Alfouzan
MB:BS, MSc, FRCPath
Associate Professor of Clinical Microbiology
Department of Microbiology
Faculty of Medicine, Kuwait University
- O. Box 24923, Safat 13110
Kuwait (+965-2498-6516)
Round 2
Reviewer 1 Report
Dear editor-in-chief
After revising all changes, the authors are responding and correcting all required changes, and answer all questions.
• Comment: The abstract and introduction are non-informative and complex; please make them more simple to read.
• Response: Changes through the abstract and introduction have been made.
• Comment: highlight all corrections and additions in red colour in the revised manuscript.
• Response: All corrections and additions have been coloured red.
• Comment:All journal names in references must be as per standard journal instructions. You can check the abbreviation according to our journal instructions.
• Response: References were rewritten according to journal instructions.
• Comment:Please add sequencing trees. Please improve your results.
Response: Sequencing trees have been added.
Author Response
Editor in Chief 6th April 2023
Diagnostics
Subject: response to reviewer #1 comments (round 2): manuscript ID: diagnostics-2308784- The Prevalence of Multi-Drug Resistant Enterobacteriaceae among Neonates in Kuwait
Thank you very much for sending us the comments of the Reviewer on the above manuscript. We have modified the manuscript in the light of comments and suggestions of the Reviewers in round 1. There were no additional comments in round 2. References have been rechecked as requested and the manuscript is uploaded.
Awaiting your positive feedback
Kind regards,
Dr. Wadha A Alfouzan
MB:BS, MSc, FRCPath
Associate Professor of Clinical Microbiology
Department of Microbiology
Faculty of Medicine, Kuwait University
P.O. Box 24923, Safat 13110
Kuwait (+965-2498-6516)